# Effects of SLC45A2 and GPNMB on Melanin Deposition Based on Transcriptome Sequencing in Chicken Feather Follicles

**DOI:** 10.3390/ani13162608

**Published:** 2023-08-12

**Authors:** Ruiting Li, Yanxing Wang, Yihan Liu, Donghua Li, Yadong Tian, Xiaojun Liu, Xiangtao Kang, Zhuanjian Li

**Affiliations:** 1College of Animal Science and Technology, Henan Agricultural University, Zhengzhou 450046, China; liruiting210@163.com (R.L.); 18700258066@163.com (Y.W.); 18300683464@163.com (Y.L.); lidonghua6656@126.com (D.L.);; 2Henan Key Laboratory for Innovation and Utilization of Chicken Germplasm Resources, Zhengzhou 450046, China; 3The Shennong Laboratory, Zhengzhou 450000, China

**Keywords:** transcriptome, melanin, melanocytes, *SLC45A2*, *GPNMB*

## Abstract

**Simple Summary:**

Understanding the molecular mechanisms involved in feather color formation is important because coloration is one of the most recognizable characteristics in chickens. In this study, after transcrip-tome sequencing of the wing and neck feather follicle tissues of chickens with different plumage colors, we retrieved differentially expressed genes (DEGs) with the same trends in both the wing and neck and then identified DEGs that may be associated with melanin deposition through GO and KEGG annotation and PPI analysis. Finally, we verified that two genes in chicken melanocytes, SLC45A2 and GPNMB, promote melanocyte melanin deposition in chickens.

**Abstract:**

As an essential genetic and economic trait, chicken feather color has long been an important research topic. To further understand the mechanism of melanin deposition associated with coloration in chicken feathers, we selected feather follicle tissues from the neck and wings of chickens with differently colored feathers (yellow, sub-Columbian, and silver) for transcriptome analysis. We focused on genes that were expressed in both the wings and neck and were expressed with the same trends in breeds with two different plumage colors, specifically, *SLC45A2*, *GPNMB*, *MLPH*, *TYR*, *KIT*, *WNT11*, and *FZD1*. GO and KEGG enrichment analyses showed the DEGs were enriched in melanin-related pathways, such as tyrosine metabolic pathway and melanogenesis, and PPI analysis highlighted the genes *SLC45A2* and *GPNMB* as associated with melanin deposition. Verification experiments in chicken melanocytes demonstrated that these two genes promote melanocyte melanin deposition. These data enrich our knowledge of the mechanisms that regulate chicken feather color.

## 1. Introduction

There are a variety of color combinations and patterns in chicken plumage, making it one of the most colorful terrestrial vertebrates; therefore, these color variations have attracted interest from researchers. In addition to attracting individuals of the opposite sex, feather color helps keep chickens away from predators [1]. In addition, the color of the plumage affects the first impression consumers have when buying chickens, and so plumage color is considered an important economic trait in the industry.

The variation in plumage colors is attributed to the distribution and deposition of melanin, including eumelanin (gray or black) and pheomelanin (yellow or red), produced within melanosomes in melanocytes, and the presence of structural colors resulting from the reflection, refraction, and scattering of light within the feathers [2,3]. Studies have shown that the biosynthesis process of melanin is a series of biochemical reactions initiated by tyrosinase catalyzing tyrosine hydroxylation in the body [4]. After tyrosinase synthesis, it enters the melanosome from the Golgi apparatus, and the active transport of the membrane sends tyrosine into melanocytes and begins to synthesize melanin [5].

The deposition of melanin in chicken feathers is more affected by genetic factors than by environmental factors [6]. With the development of modern technology and the efforts of scientists, more and more causative mutations that control chicken feather color genes have been located, such as the dominant white (I), recessive white (c), spots (mo), barring (B), dark brown (DB), extended black (E), Columbia, and others [7,8,9,10,11,12]. However, the deposition of melanin pigmentation in chicken feathers is not limited to gene polymorphisms, but also influenced by the expression of various genes related to pigment formation in feather follicles during the growth period [13,14,15,16]. The transcriptome sequencing procedure is an efficient method for identifying genes that are DEGs. In recent years, several mRNAs have been shown to affect the deposition of melanin in chickens through transcriptome sequencing analysis. For example, by transcriptome sequencing of the breast muscle tissue of Xichuan black-bone chickens of the tyrosine-added group and the normal feeding group, dopachrome tautomerase (*DCT*) and endothelin receptor B subtype 2 (*EDNRB2*) were screened for effects on the deposition of eumelanin [17]. Using transcriptome sequencing to study genes related to eyelid color in chickens, it was found that premelanosome protein (*PMEL*), *DCT*, and tyrosinase (*TYR*) were expressed much more strongly in black eyelids than in light yellow eyelids (*p* < 0.05), and these coding proteins positively regulate melanin deposition [18]. RNA sequencing technology was used to study the expression profiles of mRNA in black and white back skin tissues of Muchuan black-bone chicken, and a total of eight melanin deposition-related genes: KIT proto-oncogene receptor tyrosine kinase (*KIT*), tyrosinase-related protein 1 (*TYRP1*), *DCT*, solute carrier family 45 member 2 (*SLC45A2*), *OCA2* melanosomal transmembrane protein (*OCA2*), *EDNRB2*, transient receptor potential cation channel subfamily M member 1 (*TRPM1*), and *RAB38* were identified [19]. There have also been some studies that have been conducted by transcriptome analysis of the feather follicle tissue of chickens to screen for the deposition of melanin pigmentation in chicken feathers. For example, transcriptome sequencing of feather follicles of yellow and white feathers was performed to identify the differentially expressed genes *TYRP1*, *DCT*, *PMEL*, melan-A (*MLANA*), and hematopoietic prostaglandin D synthase (*HPGDS*) associated with pheomelanin deposition [20]. High-throughput sequencing technology was used to compare the differences in the transcriptome of black and white chicken feather bulbs, revealing the differential genes homeobox B9 (*HOXB9*), homeobox C8 (*HOXC8*), homeobox A9 (*HOXA9*), ChaC glutathione specific gamma-glutamylcyclotransferase 1 (*CHAC1*), and glutathione peroxidase 3 (*GPX3*) involved in the melanin production of black-bone chicken feathers [21]. Sub-Columbian chicken dorsal neck black and white striped feather follicles and abdominal neck white feather follicles were selected for transcriptome sequencing, and it was found that mediator complex subunit 23 (*MED23*) and G protein subunit alpha q (*GNAQ*) played a crucial role in melanin deposition [22]. However, the formation of feather color in many chicken breeds is the result of a combination of eumelanin and pheomelanin, so it is necessary to study the change in total melanin. At the same time, few transcriptome-screened genes are functionally verified in chicken melanocytes.

The “Yufen I” hybrid lines of Chinese commercial laying hens, developed by Henan Agricultural University, consist of three lines. Among these, the H line exhibits predominantly white plumage with black bars on the hackles, primaries, secondaries, and tail. This coloration closely resembles the Columbian plumage pattern, except that it features barring instead of black vertical striping [23]. Hence, we call this plumage coloration “sub-Columbian” (Figure 1A). In this experiment, we collected the follicle tissue of the wing and neck of chickens with three different plumage colors: yellow feathers (Y group) (Figure 1B), sub-Columbian feathers (H group), and silver feathers (S group) (Figure 1C). Then, we identified *SLC45A2* and glycoprotein nmb (*GPNMB*) from the differentially expressed genes revealed by transcriptome sequencing. By constructing an overexpression vector, we confirmed that these two genes promote the deposition of melanin in chicken melanocytes. Overall, our study provided a theoretical basis for the molecular mechanism underlying the formation of chicken feather color and revealed the roles of *SLC45A2* and *GPNMB* in regulation of melanin deposition during feather color growth.

## 2. Materials and Methods

### 2.1. Ethics Statement

In accordance with the protocol approved by the Institutional Animal Care and Use Committee (IACUC) of Henan Agricultural University, China (Permit Number: 11-0085; date: 13 June 2011), all sample collection and treatment procedures were carried out with strict adherence to ethical guidelines to ensure animal welfare and minimize suffering.

### 2.2. Laboratory Animal Samples and Collection

Hens with different plumage colors were selected as experimental subjects (H line chicken: sub-Columbian feathers; Gushi chicken: yellow feathers; Hy-Line chicken: silver feathers). Chickens were housed and raised in our laboratory using standard methods. The tissues were categorized into six groups: HN (sub-Columbian plumage nuchal follicle tissue), HW (sub-Columbian plumage wing follicle tissue), YN (yellow plumage nuchal follicle tissue), YW (yellow plumage wing follicle tissue), SN (white plumage nuchal follicle tissue), and SW (white plumage wing follicle tissue). Each group consisted of three biological replicates, and each replicate contained three follicle tissue samples.

Nine 5-week-old hens were selected for each group, and nuchal follicle tissue and wing follicle tissue were collected immediately. First, the surface of the feather was cleaned with 75% alcohol cotton, the feather root was clamped with tweezers, and the feather follicles were quickly removed; then, another small forceps was used to separate the feather follicle tissue at the root of the feather. The feather was cleaned with PBS, quickly placed in a cryopreservation tube, and finally frozen in a liquid nitrogen tank.

### 2.3. RNA Extraction, Library Construction, and High-Throughput Sequencing

RNA was extracted from the follicles was performed using TRIzol Reagent (Invitrogen, Carlsbad, CA, USA). The RNA quantity and quality were assessed through 1% agarose gel electrophoresis. Eighteen RNA sequence libraries were constructed—HW1, HW2, HW3, HN1, HN2, HN3, YW1, YW2, YW3, YN1, YN2, YN3, SW1, SW2, SW3, SN1, SN2, and SN3. The sequencing libraries for the NEBNext^®^ UltraTM Directional RNA Library Prep KIT (New England Biolabs, Ipswich, MA, USA) were generated on the Illumina platform. Initially, the raw data in fastq format were processed using an in-house Perl script. During this step, the raw data were filtered to obtain clean reads by excluding those containing adapters, poly-N sequences, and low-quality reads. The clean reads were mapped to the *Gallus_gallus-6.0* genome (http://ftp.ensembl.org/pub/release-96/fasta/gallus_gallus/dna/) URL (accessed on 11 March 2019) using HISAT2 [24], and the expression levels of each gene were calculated using fragments per kilobase of transcript per million mapped fragments (FPKM) [25]. The formula is shown as follows: FPKM= $\frac{cDNAFragment}{MappedFragment(Millions)*TranscriptLength(kb)}$ [25]. Differential expression analysis of the 6 groups was performed using DESeq2 [4]. To control the false discovery rate (FDR), only genes with *p* adjusted < 0.05 were considered differentially expressed.

To screen for alternative splicing events of DEGs associated with melanin deposition, equal amounts of RNA were collected from the nuchal and wing of sub-Colombian and yellow feathers into four mixing pools: the HN group, HW group, YN group, and YW group. The four libraries were then analyzed for full-length transcriptomes using the PacBio Iso-seq platform.

### 2.4. Quantitative Real-Time PCR (qRT–PCR)

For qPCR analysis, nine DEGs were chosen, including collagen type I alpha 1 chain (*COL1A1*), aquaporin 1 (*AQP1*), cysteine rich angiogenic inducer 61 (*CYR61*), regucalcin (*RGN*), keratin 23 (*KRT23*), *SLC45A2*, *GPNMB*, InaF motif containing 2 (*INAFM2*), and NFKB inhibitor epsilon (*NFKBIE*). Three biological replicates and three technical replicates were performed for each group. The gene expression levels were measured using quantitative real-time PCR (qRT-PCR) with SYBR Premix Ex TaqTM II (TaKaRa) on a LightCycler 96 Instrument (Roche Applied Science, Indianapolis, IN, USA). Glyceraldehyde-3-phosphate dehydrogenase (*GAPDH*) was used as the internal reference gene. Primers were designed using Oligo 6.0 software. The primer sequences can be found in Appendix A.

### 2.5. GO and KEGG Enrichment Analysis

Gene Ontology (GO) enrichment analysis of the DEGs was conducted with the GOseq R package (https://www.rdocumentation.org/packages/goseq/versions/1.24.0/topics/goseq, accessed on 23 June 2023). This analysis employed Wallenius noncentral hypergeometric distribution and adjusted for length bias in the DEGs [26]. The Kyoto Encyclopedia of Genes and Genomes (KEGG) is a database resource that provides insights into the advanced functions and utilities of biological systems according to the pathways to which they belong (http://www.genome.jp/kegg/). To evaluate the statistical enrichment of DEGs in KEGG pathways, we utilized the KOBAS software. Pathways with q values < 0.05 for either GO or KEGG were regarded as significantly enriched.

### 2.6. Protein–Protein Interaction Analysis

To examine the interactive connections between DEGs, we obtained protein–protein interaction (PPI) data for the DEGs from the STRING database (http://string-db.org, Organism: Chicken). Subsequently, the DEG PPI network was constructed and visualized using Cytoscape version 3.6.1 (http://www.cytoscape.org/).

### 2.7. Plasmid Construction

To construct an overexpression plasmid for the *SLC45A2* and *GPNMB* genes, the coding DNA sequences (CDSs) of the *SLC45A2* and *GPNMB* genes were amplified from H-line chicken feather follicle cDNA by PCR using gene-specific homologous recombinant clone primers (Appendix A).

Subsequently, the coding DNA sequence (CDS) was inserted into the pcDNA3.1-EGFP vector (Invitrogen, Carlsbad, CA, USA) through homologous recombination, resulting in the creation of pcDNA3.1-SLC45A2-EGFP. The accuracy of pcDNA3.1-SLC45A2-EGFP construction was confirmed by both agarose gel electrophoresis and sequencing. The procedure for constructing pcDNA3.1-GPNMB-EGFP followed the same method as that for pcDNA3.1-SLC45A2-EGFP.

### 2.8. Cell Culture, Identification, and Transfection

Melanocytes were isolated from the peritoneum of 20-day-old Xichuan black-bone chickens. The peritoneum tissue was washed multiple times with phosphate-buffered saline (PBS) containing penicillin and streptomycin. Next, the tissue was cut into pieces and digested with dispase II (Roche, China) and trypsin-EDTA (Gibco, Waltham, MA, USA) at 37 °C for 1 h with gentle shaking every 15 min. To stop the digestion, Medium 254 (Gibco, Waltham, MA, USA) supplemented with 1% Human Melanocyte Growth Supplement (HMGS; Gibco; Waltham, MA, USA) was added at a 1:3 ratio. The resulting cell suspension was then filtered through screens of varying mesh sizes (100, 200, and 500). Next, the filtered suspension was centrifuged at 1000 rpm for 5 min. The obtained cell pellet was resuspended to obtain melanocytes. These melanocytes were cultured at 37 °C and 5% CO_2_ [27].

Immunofluorescence analysis was used to confirm that isolated primary cells were melanocytes. Primary cells were fixed with 4% paraformaldehyde for 10 min and then permeabilized with 0.1% Triton X-100 for 1 h, followed by 3 washes with PBS. Then, 3% bovine serum albumin (BSA) was added for 1 h to block nonspecific binding, and melanocyte inducing transcription factor-melanocyte inducing transcription factor (MITF; 1:200, ab122982) (Abcam, Cambridge, UK) was added and incubated at 4 °C for 16 h, followed by 3 washes with PBS. The secondary antibodies used for staining were goat anti-mouse IgG conjugated to Alexa^®^ Fluor 488 (ab150078; Abcam) and goat anti-rabbit IgG conjugated to Alexa Fluor 555 (ab150117; Abcam). The samples were then incubated with the secondary antibody for 1 h in the dark and washed 3 times with PBS. The cells were incubated with DAPI (Abcam) for 10 min at room temperature in the dark and quickly washed 3 times with PBS to stain the nuclei. Finally, the cells were observed under an inverted fluorescence microscope.

Transient transfections were carried out using Lipofectamine 3000 (Invitrogen, Carlsbad, CA, USA) according to the manufacturer’s instructions, and the cells were collected after 48 h of incubation.

### 2.9. Western Blotting (WB)

The proteins were extracted from the cells and analyzed by western blotting. The protein expression of MITF and GAPDH was examined through western blot analysis. Primary antibodies, including anti-MITF (MITF; 1:1000, bs-1990R) (Bioss, Beijing, China) and anti-GAPDH (GAPDH; 1:10,000; AF7021) (Affinity, West Bridgeford, UK), were used in this study.

### 2.10. Tyrosinase and Melanin Content Measurements

After transfection for 48 h, the melanocytes were washed twice with ice-cold PBS, digested with trypsin and centrifuged for 10 min to obtain the pellet. Then, intracellular tyrosinase was measured by enzyme-linked immunosorbent assay (ELISA) using a Tyrosinase ELISA Assay KIT (Nanjing Jiancheng Bioengineering Institute) according to the manufacturer’s instructions. Melanin content was quantitated using the Amplite™ Fluorimetric Melanin Assay Kit (ATT Bioquest, CA, USA).

### 2.11. Statistical Analyses

Statistical analysis was conducted using SPSS Statistics 27 to assess differences among groups. One-way ANOVA was performed, and significance levels were indicated by * (*p* < 0.05) and ** (*p* < 0.01). The data are presented as the mean ± SEM (standard error of the mean).

## 3. Results

### 3.1. Transcriptome Sequencing and Quality Control

Eighteen libraries were sequenced on an Illumina HiSeq platform. As shown in Appendix A, clean reads, mapped reads, uniquely mapped reads, and Q20, Q30, and GC contents for each library were identified. A total of 476.78 Mb of clean data was obtained, with an average of 26.49 Mb of clean data for each sample. Of these, the uniquely mapped reads to the chicken reference genome (*Gallus_gallus-6.0*) ranged from 77.06% to 84.86%. The sequencing data obtained from the experiment exhibited high quality, as indicated by Q20 (>95%) and Q30 (>90%) values and the similar GC content. To ensure accessibility and transparency, the raw sequence data have been deposited in the NCBI Sequence Read Archive under accession number PRJNA859098.

### 3.2. Differential mRNA Expression in Feather Follicles

DESeq2 was used to analyze the DEGs between the groups. Fold change ≥ 1.5 and FDR < 0.05 were used as screening criteria for DEG detection. A total of 12104 genes were detected in this study; 276 DEGs were obtained from the YW group and the HW group, of which 149 genes were upregulated and 127 genes were downregulated; and 212 DEGs were obtained from the YN group and the HN group, of which 98 genes were upregulated and 114 genes were downregulated (Figure 2A,B). Compared to the SW group, the HW group had 4198 DEGs identified, with 2591 genes upregulated and 1607 genes downregulated. A total of 7418 DEGs were obtained in the SN group and the HN group, including 4334 upregulated and 3084 downregulated genes (Figure 2C,D). To identify the DEGs responsible for the formation of plumage color, we targeted DEGs that had the same trend in the YW vs. HW groups and in the YN vs. HN groups or that had the same trend in the SW vs. HW groups and in the SN vs. HN groups (*p* < 0.05) (Figure 2E). If a DEG had the same trend in the YN vs. HN group and the YW vs. HW group, we denoted it as the Y vs. H group and similarly for the S vs. H group.

To some extent, alternative splicing allows the same gene to produce multiple isoforms. Four AS forms were detected, namely, the alternative 3′ splice site (A3SS), alternative 5′ splice site (A5SS), retained intron (RI), and skipped exon (SE) [28]. We identified AS forms of genes expressed in feather follicle tissues of the HN vs. YN group and HW vs. YW group, which showed similar proportions for the two groups. Unfortunately, we did not find different alternative splicing events in the DEGs of the HN vs. YN groups or in the HW vs. YW groups.

### 3.3. Data Validation

We randomly selected nine DEGs for validation using qRT-PCR and found that the expression levels of all genes were consistent with the results of RNA-seq, indicating that the detection and expression of our RNA-seq are accurate (Figure 3).

### 3.4. GO Analysis and KEGG Pathway Enrichment Analysis

Gene enrichment analysis was conducted to investigate the distribution of GO items and reveal differential enrichment in the biological process, cellular component, and molecular function categories. The GO classification of the DEGs can be observed in Figure 4 and Appendix A. Within the biological process category, the predominant subcategories identified in all four groups included cellular process, single-organism process, biological regulation, and metabolic process.

The cellular process was the most affected of the subcategories, including upregulated genes *AQP1* in the Y vs. H groups, ras homolog family member C (*RHOC*), frizzled class receptor 1 (*FZD1*), frizzled class receptor 6 (*FOXO6*), dihydroorotate dehydrogenase (quinone) (*DHODH*) in the S vs. H groups, downregulated genes *TYR*, *KIT*, transient receptor potential cation channel subfamily M member 2 (*TRPM2*), *GPNMB* in the Y vs. H groups, Wnt family member 11 (*WNT11*) in the S vs. H groups, etc. Among the cellular components, the cell, cell part, and organelle categories were dominant. The DEGs involved in these GO terms included the upregulated genes *DHODH*, *FZD1*, myocardin (*MYOCD*), LIM domain kinase 1 (*LIMK1*), and semaphorin 6B (*SEMA6B*) in the S vs. H groups and the downregulated genes *SLC45A2*, *TYR*, *MLPH*, major facilitator superfamily domain containing 12 (*MFSD12*), and *TRPM2* in the Y vs. H groups. For molecular function, the most affected subcategories were bound together (upregulated genes included aquaporin 1 (*AQP1*) and *CYR61* in the Y vs. H groups, RHOC, SRY-box 9 (*SOX9*), *FZD1*, and forkhead box O6 (FOXO6) in the S vs. H groups, and downregulated genes included *TYR*, *MLPH*, and *GPNMB* in the Y vs. H groups, and WNT11 in the S vs. H groups), catalytic activity (upregulated genes included LIM domain kinase 1 (*LIMK1*), *RHOC*, alkaline ceramidase 1 (*ACER1*), protein phosphatase, Mg2+/Mn2+ dependent 1J (*PPM1J*), and aprataxin (*APTX*) in the S vs. H groups, and downregulated genes included *TYR*, *KIT*, and *TRPM2* in the Y vs. H groups), and transporter activity (upregulated genes included gap junction protein beta 1 (*GJB1*), tRNA-yW synthesizing protein 5 (*TYW5*), cyclin, and CBS domain divalent metal cation transport mediator 1 (*CNNM1*), and solute carrier family 11 member 1 (*SLC11A1*) in the S vs. H groups, and downregulated genes included *SLC45A2* in the Y vs. H groups and glutamate ionotropic receptor kainate type subunit 1 (*GRIK1*) in the S vs. H groups. These genes play an important role in the deposition of melanin.

The results of KEGG pathway enrichment analysis for differentially expressed genes are shown (Figure 5, Appendix A), listing the top 20 pathways with the lowest Q values. Among the pathways that are significantly associated with melanin are tyrosine metabolism, the MAPK signaling pathway, adrenergic signaling in cardiomyocytes, and the calcium signaling pathway. Among these pathways, the tyrosine metabolism pathway was the most significantly enriched pathway in the Y vs. H groups (*p* < 0.05). By analyzing the top 20 pathways with the lowest Q values in the four comparison groups, we found the distinctly expressed gene *TYR* in the tyrosine metabolic pathway.

In addition, the KEGG classification table shows that there were other melanin-related pathways, such as melanogenesis, Wnt signaling, Notch signaling, calcium signaling, adherens junction, and mTOR signaling in the four comparison groups (Appendix A). Interestingly, we found that the melanogenesis pathway was common in all four groups, so we focused on that pathway (Figure 6). We found that nine DEGs were enriched in the melanogenesis pathway—*TYR*, *KIT*, *WNT11*, *FZD1*, frizzled class receptor 2 (*FZD2*), frizzled class receptor 7 (*FZD7*), frizzled class receptor 9 (*FZD9*), adenylate cyclase 3 (*ADCY3*), and adenylate cyclase 3 (*ADCY8*), which had the same trend in the YW vs. HW groups and the YN vs. HN groups, or had the same trend in the SW vs. HW groups and SN vs. HN group (Appendix A).

### 3.5. PPI Analysis of DEGs

In the Y vs. H groups, the DEG PPI network was composed of nine proteins and nine pairs of PPIs. Moreover, *TYR* interacted with *SLC45A2*, *KIT*, *GPNMB*, and *MFSD12.* Melanophilin (*MLPH*) interacted with *SLC45A2*, *TYR*, and *MFSD12* (Figure 7A). In addition, *TYR* was mainly enriched in tyrosine metabolism and melanogenesis. Furthermore, the PPI network in the S vs. H groups consisted of 161 proteins and 107 pairs of PPIs (Appendix A). The key core node *FZD1* exceeded seven interactions, including *WNT11*, transcription factor 7-like 2 (*TCF7L2*), and *RHOC*. Among them, both heat shock protein family A (Hsp70) member 8 (*HSPA8*) and heat shock protein family B (small) member 1 (*HSPB1*) were enriched in the MAPK signaling pathway, while *WNT11* and *TCF7L2* were enriched in the melanogenesis pathway and the Wnt signaling pathway (Figure 7B).

### 3.6. Effects of SLC45A2 and GPNMB Overexpression on Melanin Deposition

Notably, we found that the expression of *SLC45A2* and *GPNMB* in the H line was significantly lower than that in the Gushi line using transcriptome sequencing and qRT–PCR. Moreover, in the transcriptome sequencing, the eumelanin-related genes *MLPH* and *TYR* were significantly reduced in Group H compared to Group Y (*p* < 0.05). Given that *SLC45A2* has the same expression trend as *MLPH* and *TYR*, high expression of *SLC45A2* and *GPNMB* is also speculated to affect the deposition of melanin.

To explore the biological role of *SLC45A2* and *GPNMB* in eumelanin in chickens, we overexpressed the two genes separately in chicken primary melanocytes. Primary melanocytes were identified by immunofluorescence staining for MITF (Figure 8A). The mRNA expression levels of the *SLC45A2* and *GPNMB* genes in melanocytes were increased by approximately 18000-fold and 3-fold after transfection with pcDNA3.1-SLC45A2-EGFP and pcDNA3.1-GPNMB-EGFP, respectively, compared to the control group that was transfected with the pcDNA3.1-EGFP plasmid (Figure 8 B,C).

After overexpression of *SLC45A2* or *GPNMB*, the mRNA expression levels of the eumelanin marker genes *TYR*, *MLPH*, and *MITF* were significantly increased (*p* < 0.01; Figure 8D,E). Consistently, western blotting also confirmed that the protein levels of MITF were upregulated following melanocytes’ transfection with pcDNA3.1-SLC45A2-EGFP and pcDNA3.1-GPNMB-EGFP, respectively, compared to the control group (Figure 8F,G). Full WB images can be found in Appendix A. Furthermore, the tyrosinase and melanin contents in cells transfected with the recombinant plasmids pcDNA3.1-SLC45A2-EGFP or pcDNA3.1-GPNMB-EGFP were significantly higher than those in the control group (*p* < 0.01) (Figure 8H–K).

## 4. Discussion

In chicken feathers, melanin distribution and deposition are responsible for plumage color variations, and feather follicles are the only place where plumage melanin is produced [29]. During feather follicle melanogenesis, mature melanocytes produce melanosomes; newly synthesized melanin then migrates to the feather keratinocytes, giving the feathers their particular color [30]. Pigmentation is a complex feature, and many studies have shown that melanin pigmentation is strongly genetically controlled [31]. In recent years, several key genes involved in melanin deposition have been identified. These include *MLPH*, *TYR*, *KIT*, *PMEL*, *MITF*, agouti signaling (*ASIP*), melanocortin 1 receptor (*MC1R*), and endothelin 3 (*EDN3*) [32]. However, there has been little research on the association of pigmentation on chicken feathers. In particular, because most of the feather color is caused by eumelanin and pheomelanin, and most of the marker genes related to melanin deposition, for example *TYR*, affect both eumelanin and pheomelanin, it is particularly important to further study the colorful plume color due to total melanin deposition [33,34]. In our transcriptome data, there were significant differences in the expression of *MLPH*, *TYR*, and *KIT* (*p* < 0.05). *MLPH* is an important structural protein that assists in the transport of melanosomes, ensuring that melanosome bodies accumulate at the dendritic borders of melanocytes and release them into surrounding tissues for deposition, regulating the color of the animal’s skin and coat. *MLPH* gene mutations result in chickens with the Japanese quail feather lavender diluted phenotype [35,36]. The expression of the *MLPH* gene in the goose has tissue specificity, with the highest expression in black eyes and the lowest expression in back skin, abdominal skin, and fins, but it was not detected in the heart, liver, and other organs [37]. This expression pattern is consistent with the melanin deposition pattern of geese, showing that the *MLPH* gene is related to melanin deposition [37]. *TYR* is a key gene in the biosynthesis of melanin in animals, and the tyrosinase it encodes has the catalytic activity of tyrosine hydroxylase and dopa oxidase and is a rate-limiting enzyme in the process of melanin biosynthesis. The lack of tyrosinase catalytic function hinders melanin synthesis, resulting in corresponding albinism traits in the animal’s coat and skin [38]. At present, many research results show that functional mutations in the *TYR* gene and differences in gene expression levels in animals such as cattle, sheep, dogs, cats, martens, rabbits, rats, and poultry will lead to albino traits in their coats [14,39,40]. The insertion of an intact avian retroviral sequence in intron 4 in the chicken *TYR* gene causes deletion of transcript exon 5, eventually leading to chicken recessive white feather mutation [8]. The mast stem cell factor receptor encoded by the *KIT* gene is a member of the tyrosine kinase receptor family, which acts on blood cells and melanin, affecting the growth and development of blood cells and melanocytes [41]. When *KIT* is expressed in a melanocyte precursor, its ligand SCF activates receptors on the cell membrane through the MAPK signaling pathway, ultimately initiating melanin synthesis [42]. Together with the previous results, we speculate that the expression of *MLPH*, *TYR*, and *KIT* plays an important role in the deposition of melanin in chicken feathers.

To understand the potential functions of the identified mRNAs, the DEGs were subjected to functional enrichment analyses. KEGG pathway analysis showed that the differentially expressed genes *TYR*, *KIT*, *WNT11*, *FZD1*, *FZD2*, *FZD7*, *FZD9*, *ADCY3*, and *ADCY8* were significantly enriched in tyrosine metabolic pathway or melanogenesis. This result suggested that the DEGs identified in the feather follicles play an important role in melanin biosynthesis and pigmentation. In previous studies, we collected dorsal feather follicles (sub-Columbian feathers) and ventral feather follicles (white feathers) of the nuchal area of the H line for RNA-seq and found that the expression of *WNT11* and *SLC45A2* significantly differed in these two sites, while *WNT11* was enriched in the melanogenesis pathway [22]. *WNT11* also plays a crucial role in the Wnt signaling pathway. Several studies have demonstrated that the Wnt pathway is involved in melanin synthesis [43,44]. In addition, we found that the expression of *WNT11* in the feather follicles of the H line was significantly increased after feeding 1.0% tyrosine daily [22]. However, there are very few reports on the synthesis of *WNT11* in melanin. In this study, the expression level of *WNT11* in sub-Columbian feathers was significantly higher than that in white feathers (*p* < 0.05). Using KEGG enrichment analysis, we found that *WNT11* was enriched in the melanogenesis pathway and Wnt signaling pathway. In the PPI network, *WNT11* interacted with *FZD1*, which is also involved in the melanogenesis pathway. We speculate that *FZD1* and *WNT11* are responsible for the formation of dark spots (melanin) at specific sites in sub-Columbian feathers, but the exact genetic mechanism needs to be further investigated.

In our transcriptome data, we focused on DEGs (*p* < 0.05) between different plumage colors and selected DEGs with the same trend in both wing and neck. Moreover, through PPI analysis, genes that are consistent with the expression trend of *MLPH*, *TYR*, and *KIT*, such as *SLC45A2* and *GPNMB*, are likely to exhibit similar functions to melanin marker genes. 

*SLC45A2* is often found to be overexpressed in black tissues compared with white tissues [45]. Moreover, after transfection of *SLC45A2* into mouse melanocytes, *MITF*, *TYR*, *TYRP1*, *DCT*, and tyrosine contents were significantly increased (*p* < 0.05) [46]. This is similar to the result of significantly elevated *MITF* and *TYR* after we overexpressed *SLC45A2* on chicken primary melanocytes (*p* < 0.05). According to the results of previous experiments in our laboratory, the expression of *SLC45A2* on the dorsal side of the nuchal (sub-Columbian plumage) area of the H line was significantly higher than that on the ventral side of the nuchal (white plumage) area, and the addition of different concentrations of tyrosine to melanocytes significantly affected the expression of *SLC45A2* [22]. In our experimental results, after overexpressing *SLC45A2*, the tyrosinase content and melanin content in cells were significantly increased, so we hypothesized that the high expression of *SLC45A2* could promote the deposition of melanin. Notably, *SLC45A2* is the known pathogenic gene of an autosomal recessive hypopigmentary disorder, oculocutaneous albinism type 4 (OCA4) [45]. Moreover, the missense mutation of *SLC45A2* results in a dominant silver locus that dilutes pheomelanin [47,48]. Similarly, no effect on eumelanin was observed in *SLC45A2* mutant horses, with some effect on pheomelanin dilution [49]. Cysteine is a key amino acid in the synthesis of pheomelanin, so we speculate that this mutation prevents cysteine transport to melanosomes. In addition, *SLC45A2* transports sugar in yeast cells in an acid-dependent manner [50]. The anti-melanogenic effects of sugar and sugar derivatives are probably due to three different mechanisms: by increasing melanosomal pH, by interfering with melanosome maturation, or by inhibiting TYR maturation by blocking N-glycosylation [51]. It has been previously demonstrated that the proton/glucose exporter *SLC45A2* mediates melanin synthesis and melanoma metastasis primarily by modulating melanosomal glucose metabolism. In human melanocytes and melanoma cell lines, the *SLC45A2* gene is highly enriched, and MATP, the gene’s protein, is found in melanosomes [52]. By knocking down MATP using siRNAs, melanin content and tyrosinase activity were reduced. It helps maintain an appropriate melanosome pH, so that copper ions can be incorporated into the tyrosinase active site correctly [52]. Therefore, we speculate that in the melanocytes of chicken feather follicles, *SLC45A2* may regulate the pH of melanosomes by regulating the glucose metabolism of melanosomes, leading to the deposition of melanin.

At the cyclin-dependent kinase inhibitor 2A (*CDKN2A*) tumor suppressor locus, there are two non-coding and two missense mutations associated with sex-linked barring [53]. Schwochow Thalmann found that one or both of the non-coding changes cause an upregulation of *CDKN2A* expression in feather follicles during feather growth [10]. Melanocyte progenitor cells may stop dividing prematurely when *CDKN2A* expression is high, distinguishing into pigment-producing cells instead [10]. In the KASP typing results published earlier in our laboratory, the silver feather only had the mutations of *SLC45A2* and no mutation of cyclin-dependent kinase inhibitor 2A (*CDKN2A*), while sub-Columbian plumage have both *SLC45A2* mutations and *CDKN2A* mutations. We speculate that *CDKN2A* is also a key gene influencing melanin deposition in chicken feathers. But *CDKN2A* did not appear in the differentially expressed genes in our transcriptome data. However, we verified that the gene was significantly different in the sub-Columbian plumage vs. yellow plumage by q-PCR (*p* < 0.05) (Appendix A). The results were consistent with the previous studies.

Melanin is synthesized in the melanosomes of melanocytes, and its formation goes through four stages [54]. GPNMB is a type I transmembrane glycoprotein, which is present in all stages of melanosome formation (I–IV), and especially enriched in mature (stage III and IV) melanosomes [55,56]. The expression level of *GPNMB* in melanocytes was found to be inversely correlated with the metastatic capacity of human melanomas and was shown to be associated with the development of the retinal pigment epithelium and the iris [57,58]. In mice with pigmentary glaucoma, a premature stop codon mutation in the GPNMB gene results in iris pigment dispersal [59]. Combined with the results of this experiment, in chicken melanocytes, overexpression of *GPNMB* can increase intracellular tyrosinase content and melanin content, we speculated that the high expression of *GPNMB* could promote the deposition of melanin. During development, *GPNMB* is expressed in a pattern similar to that of *MITF*, *DCT*, and *PMEL* [60]. *GPNMB* and *PME*L are highly homologous proteins that play an important role in melanosome biogenesis [61,62]. In addition, *GPNMB* adheres to *PAM212* keratinized cells in an RGD-dependent manner and is possibly involved in melanocyte development, melanin synthesis, and melanoma production [63]. *MITF* has been regarded as an essential transcription factor for the regulation of more than 25 pigmentation genes, such as *TYR*, *TYRP1*, *DCT*, *PMEL17*, and so on [64]. In addition, *MITF* is involved in the Wnt/β-catenin signaling pathway, MC1R/α-MSH signaling pathway, and SCF/C-KIT signaling pathway [65,66,67]. The first two pathways can increase *MITF* expression, and these pathways affect melanin production by regulating the transcription of *TYR*, *TYRP1*, and *DCT* through *MITF* [68]. Based on this experiment, after overexpression of *GPNMB* in chicken melanocytes, the expression of *TYR* and *MITF* increased. From this, we speculate that *GPNMB* affects the deposition of melanin by upregulating the transcription of *TYR*, *TYRP1*, and *DCT* by upregulating *MITF*.

It is worth noting that by screening for differentially expressed genes with the same expression trend as marker genes such as *TYR* in the two groups of this experiment, combined with PPI correlation analysis, we finally speculated that *SLC45A2* and *GPNMB* genes may affect the deposition of total melanin on chicken feather follicles. However, since the experimental cells were obtained from the peritoneal tissue of black chickens, the cells can only secrete eumelanin, so we can only prove that *SLC45A2* and *GPNMB* have an effect on the deposition of eumelanin, and whether they have an effect on pheomelanin needs further experimental verification.

## 5. Conclusions

In this study, transcriptome sequencing and subsequent GO, KEGG, and PPI analyses were used to screen for genes with the same trend as melanin deposition marker genes and DEGs (*SLC45A2*, *GPNMB*, *MLPH*, *TYR*, *KIT*, *WNT11*, and *FZD1*) in the neck and wing feather follicles of chickens. Moreover, we found that *SLC45A2* and *GPNMB* can promote the deposition of melanin in chicken feathers. Overall, the candidate genes detected in these experiments can help provide novel insights into the mechanisms regulating feather color development in chickens.

## Figures and Tables

**Figure 1 animals-13-02608-f001:**
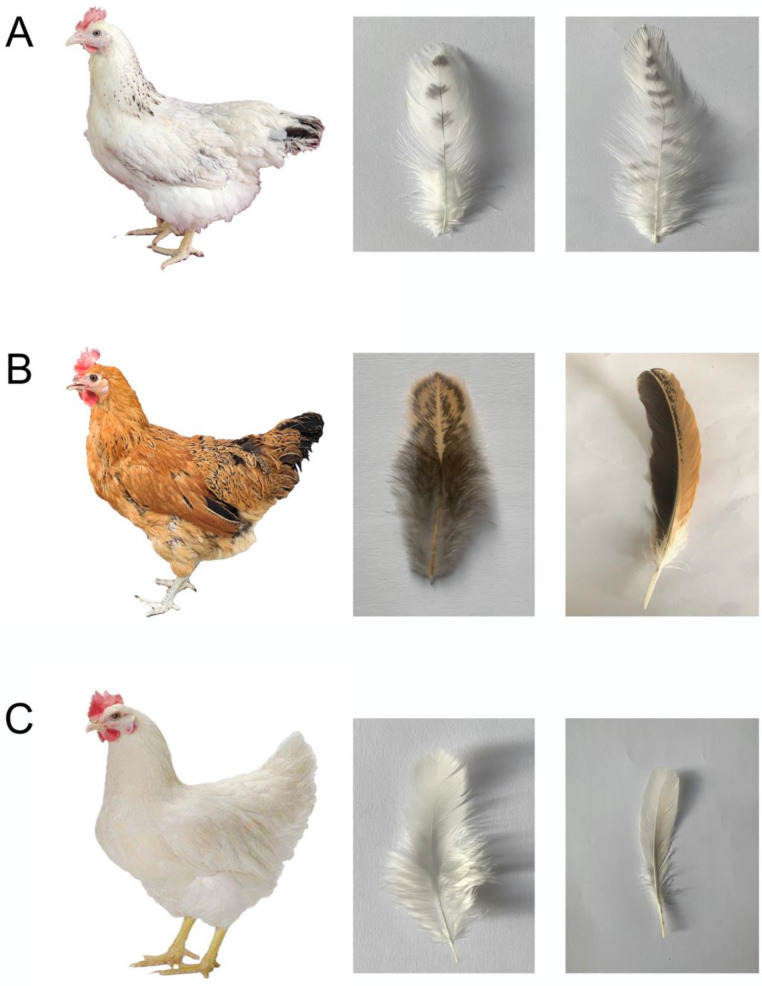
The feather colors of different colored hens, as well as their neck and wing feathers. (**A**) line chicken, sub-Columbian feathers: almost white with black barring only in the hackles, primary and secondary feathers, and tail; (**B**) Gushi chicken, yellow feathers: yellow with black spots all over the body feathers; (**C**) Hy-Line chicken, silver feathers: pure white feathers all over the body.

**Figure 2 animals-13-02608-f002:**
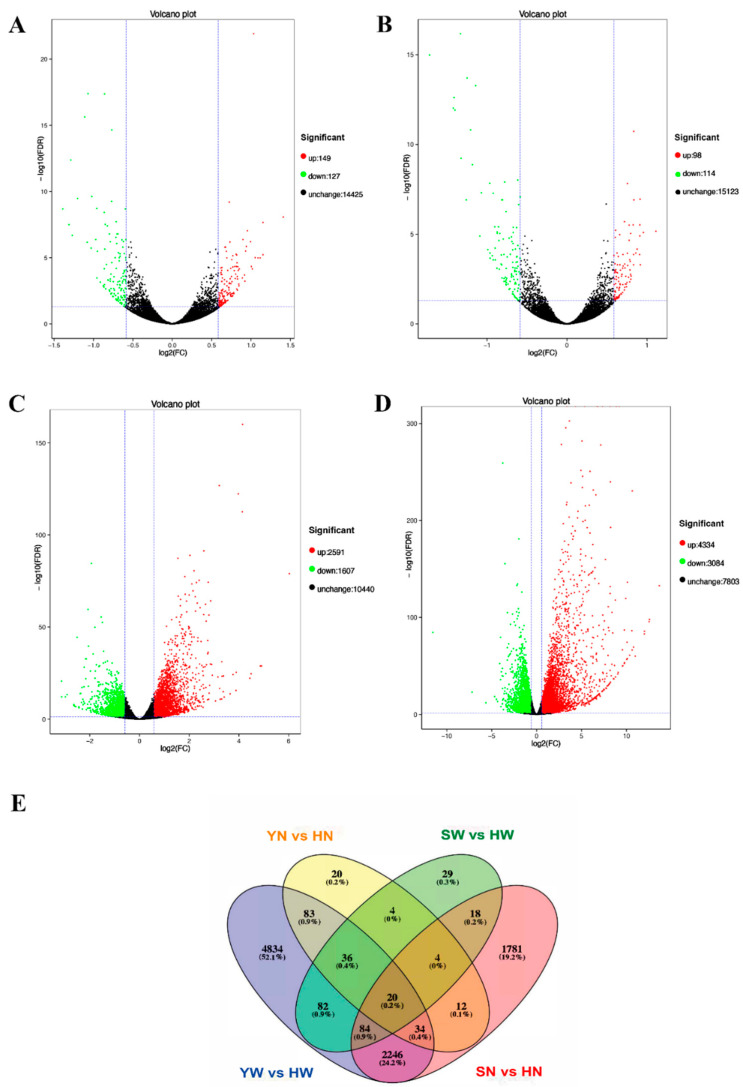
Analyses of DEGs in four comparisons. (**A**–**D**) YW vs. HW, YN vs. HN, SW vs. HW, and SN vs. HN groups, respectively. Red and green points indicate the genes with significantly increased or decreased expression, respectively (FDR < 0.05). The x-axis shows the log2-fold change in expression, and the y-axis shows the log10-fold likelihood of a gene being differentially expressed. (**E**) Venn diagram showing the overlap of the DEGs in four comparisons.

**Figure 3 animals-13-02608-f003:**
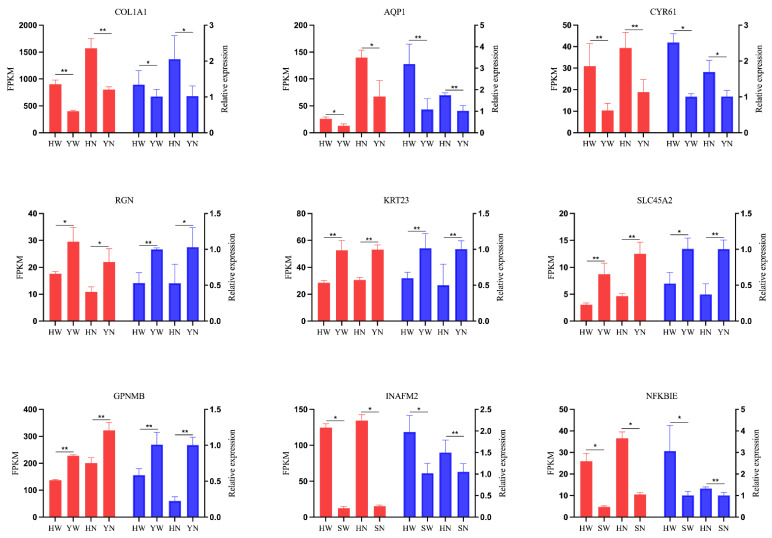
RT–qPCR validation of RNA-seq results. The left Y-axis displays the FPKM derived from the RNA-seq, while the data from RT–qPCR are shown on the Y-axis on the right. The data are represented as the means ± SEDs; red represents the FPKM value of RNA-seq, blue represents the RT–qPCR with GAPDH as the internal reference gene; * indicates *p* < 0.05; ** indicates *p* < 0.01.

**Figure 4 animals-13-02608-f004:**
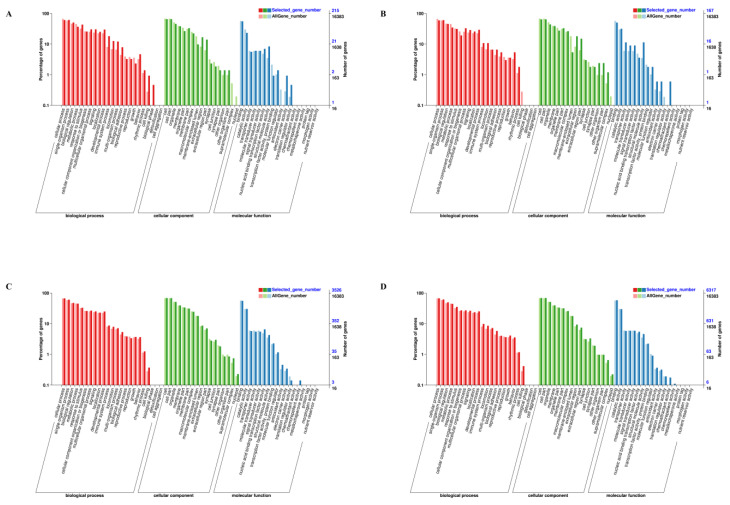
GO classification of DEGs. (**A**) YW vs. HW groups. (**B**) YN vs. HN groups. (**C**) SW vs. HW groups. (**D**) SN vs. HN groups. The x-axis indicates the secondary classification in the GO database; the y-axis on the left side indicates the ratio of the number of genes annotated with this GO classification to all genes. The y-axis on the right side indicates the number of genes annotated with this GO entry. The denominator of ‘Selected_gene_number’ is the number of DEGs, and the denominator of ‘AllGene_number’ is the total number of genes.

**Figure 5 animals-13-02608-f005:**
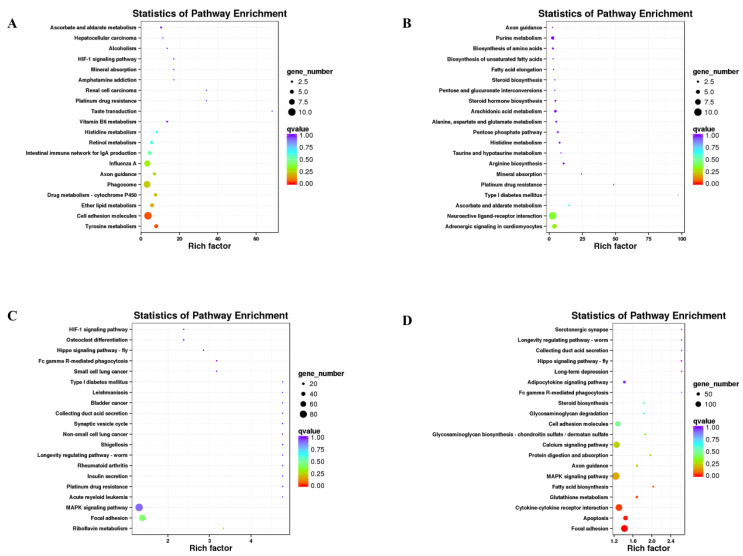
Scatterplot of the top 20 pathways in KEGG enrichment. (**A**) YW vs. HW groups. (**B**) YN vs. HN groups. (**C**) SW vs. HW groups. (**D**) SN vs. HN groups. The x-axis represents the rich factor, which is the ratio of the DEGs annotated with the pathway term to the total number of genes annotated with the pathway term. The greater the rich factor is, the greater the degree of enrichment. The y-axis shows each KEGG pathway name. Each round point represents a specific KEGG pathway. The circle size indicates the number of DEGs associated with each significantly enriched pathway. The circle color indicates the significance level (q-value). A q-value < 0.05 was considered to indicate significant enrichment. Light purple indicates the least significant, and orange represents the most significant.

**Figure 6 animals-13-02608-f006:**
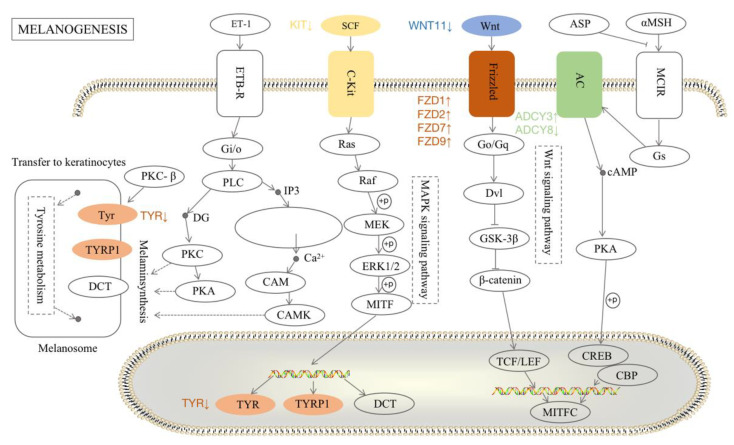
The networks of DEGs in the melanogenesis pathway. Proteins and DEGs in the same families are represented by the same color.

**Figure 7 animals-13-02608-f007:**
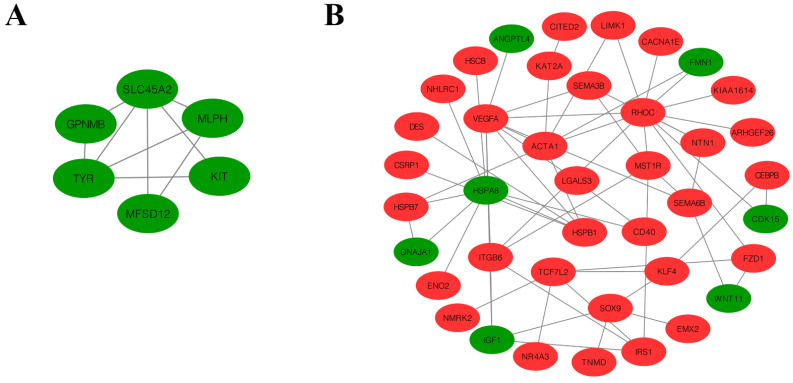
PPI network of DEGs. (**A**,**B**) represent Y vs. H and S vs. H, respectively. Red nodes represent upregulated DEGs, and green nodes represent downregulated DEGs.

**Figure 8 animals-13-02608-f008:**
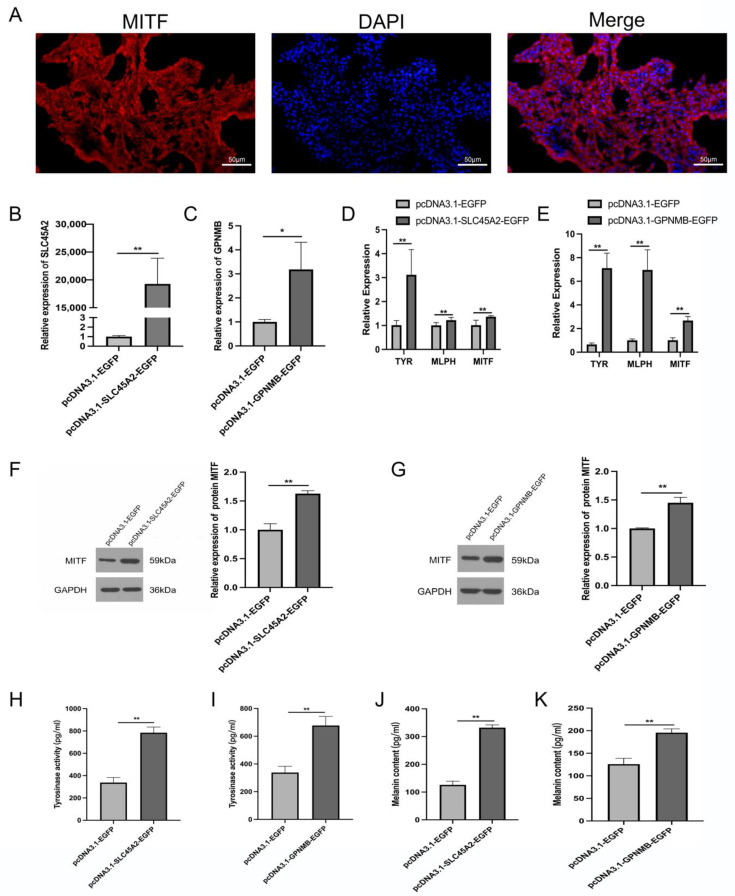
*SLC45A2* and *GPNMB* promote the deposition of melanin in chicken melanocytes. (**A**) Marker gene identification in chicken primary melanocytes (scale 50 μm). (**B**,**C**) The mRNA expression of *SLC45A2* and *GPNMB* after 48 h of transfection of the *SLC45A2* overexpression vector pcDNA3.1-SLC45A2-EGFP and the *GPNMB* overexpression vector pcDNA3.1-GPNMB-EGFP in melanocytes. (**D**,**E**) Effects of *SLC45A2* and *GPNMB* overexpression on the expression of genes related to pigmentation melanocytes. (**F**,**G**) Effects of *SLC45A2* and *GPNMB* overexpression on MITF protein levels. (**H**,**I**) Effect of *SLC45A2* and *GPNMB* overexpression on the tyrosinase content of melanocytes. (**J**,**K**) Effect of SLC45A2 and GPNMB overexpression on the melanin content of melanocytes. * *p* < 0.05; ** *p* < 0.01.

## Data Availability

The raw data used in this study are publicly available and can be obtained upon reasonable request to the corresponding author.

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
