# Peer review of "Effects of SLC45A2 and GPNMB on Melanin Deposition Based on Transcriptome Sequencing in Chicken Feather Follicles"

_animals, 2023, doi:10.3390/ani13162608_

Round 1

Reviewer 1 Report

In the manuscript titled “Effects of SLC45A2 and GPNMB on melanin deposition based on transcriptome sequencing in chicken feather follicles” Li and co-authors conducted RNA-seq for plumage color variation, using feather follicle tissues from different body regions, and from different chicken breeds. Through multiple steps of screening and filtering, they identified SLC45A2 and GPNMB that were associated with melanin deposition in the feathers. The effects of these 2 genes on chicken melanocytes were also shown via overexpression in cell culture experiments. The experimental design is straightforward and the results are well presented. Here are my comments and suggestions.

1. Lines 23-25: Since YW, HW et al. were not explained ahead of it, this sentence is not very meaningful. Even the information (how many DEGs were detected) is not a key result that should be presented in the abstract.

2. Lines 58-62: These 2 sentences are confusing, the descriptions of the "most research" in the first sentence and "few studies" in the second sentence seem similar, but they should have opposite meanings. Suggests to rewrite these 2 sentences.

3. Lines 70-71, the role of SLC45A2 in chicken plumage color is not previously unrecognized. It is responsible for the Silver mutation.

4. Lines 95, 99, 156, 164, 170: The word "KIT" should not be Italic.

5. Line 178: There is no Table 2, either Table 1.

6. Line 185: I was unable to find PRJNA859098 on NCBI, make sure that the data will be released before the publication.

7. Lines 198, 263, and 264: MW and MN were not explained. Maybe they should be YW and YN. The abbreviations in Table S4 are even more confusing.

8. Line 203: Only four AS forms were listed, not five.

9. Front sizes of all figures are too small, especially in Figures 2, 4 and 5, they can not be read at all even on the screen.

10. Line 214: It would be such a coincidence if these 9 genes were randomly selected, since 2 target genes and many melanocyte marker genes were selected in these 9.

11. Line 218: More information is needed, such as explanations of abbreviations, internal control genes, numbers of technical repetition and biological repetition, and so on.

12. Line 223: GO information is not in any of the Tables.

13. Line 268: explain "different colors".

14. Lines 295 - 310: "Figure" should be changed into "Figure 8" .

15. Figure 8: Only 8E shows the NC group, how about others in Figure 8?

16. Lines 356-357: this study was not focused on the difference between different body regions, but between breeds.

17. Lines 383 and 391: check carefully which words should be italic and which ones should not.

18. The discussion is a little bit weak, most of it is reviewing previous studies, and how they support the results from the current study. Something new could be generated in this section, such as: if the sub-columbian plumage is caused by SLC45A2 and CDKN2A mutations (based on the authors' previous studies), then the main difference in genetic between H and S group in this study should be the CDKN2A mutations. How could CDKN2A contribute to the RNA-seq results, although it is not a DEG? Then what is the difference in genetics between the H and Y groups since they both show columbian plumage? How this difference in genetics can explain the RNA-seq results. Or, SLC45A2 and GPNMB are finally identified, is there any discussion about if they affect eumelanin or pheomelanin productions, or both?

As mentioned above.

Reviewer 2 Report

The authors screened for differentially expressed genes SLC45A2, GPNMB, MLPH, TYR, KIT, WNT11 and FZD1 associated with melanin deposition by transcriptome sequencing of hair follicles of different plumage colors. They also found that SLC45A2 and GPNMB in chicken melanocytes promote the proliferation of melanocytes and the deposition of melanin in chickens, which is novelty. Besides, the sample size and the workload is enough. However, there are still some areas for revision in the article.

1、Line 84, “ (YN group)” should be followed by “;”.

2、Line 109, please replace "q adjusted" with "p-adjusted".

3、Line 135, please check the " pcDNA3.1-SLC45A2" in the sentence.

4、Please identify for chicken melanocytes.

5Line 206-208, the full-length transcriptome sequencing does not get good results, so what are you doing this for? Are there any alternative splicing events unique to the H group?

6The authors ultimately selected two genes, SLC45A2 and GPNMB for functional validation at the cellular level. What is the relationship between these two genes or proteins? If so, please add clarification.

7The pigment produced by the organism is composed of eumelanin and pheomelanin, which pigment is affected by these two genes of the author? Please add clarification.

8Genes should be indicated in italics, please unify all gene formats, such as GAPDH in line159.

English language needs native speakers to polish it.

Reviewer 3 Report

1. General comments

The manuscript, entitled as "Effects of SLC45A2 and GPNMB on melanin deposition based   on transcriptome sequencing in chicken feather follicles" report the identification of SLC45A2 and GPNMB as differentially expressed genes from comparative transcriptome analysis on the feather follicles of  of necks and wings from three breeds of chickens showing color differences, and investigate the effect of overexpression of SLC45A and GPNMB on melanin production and proliferation of chicken melanocytes.  Experimental designs are reasonable, and data interpretation is sound. This study deserves to be shared within the community. Nonetheless, improvements should be made to better acceptance by readers.

- It seems to be important to collect the hair follicles from the feather of neck and wing. There are no detailed procedures for isolation of feathers. It is recommended to add references or detailed procedures if available.

- Overexpression of two genes increased cellular proliferation and production of melanin. What would be the underlying mechanisms? If previous studies that linked to the cell proliferation and melanin production affected by these proteins, please provide the plaucible mechanims.

-Figure 1: pleases add more detail information on the chickens used in this study. 

-Figure 2:  Please improve the quality of images.

-The quality of Figure 4 and 5 should be improved. It is recommended to place them in separate pages.  In addition, add more details to legends.

- In Figure 6, more details should be provided, such as color of the genes.  

- Figure 8. Please rephrase, "SLC45A2 and GPNMB promote the production and proliferation of melanin in chicken  melanocytes."  SLC45A2 and GPNMB promote the production of melanin and proliferation of chicken melanocytes?

2. Specific comments

Line 24-25. Please provide full names of SW, HW, SN, and HN.

Line 71. Please include the full name of SLC45A2 and GPNM, because the these genes appear for the first time in the main text.

Line 92. The contamination of RNA was detected using 1% agarose gel electrophoresis --> Please make sure whether "contamination of RNA" or "contamination of DNA".  

Line 96.  Please rephrase, "the total dose per sample was 1 μg RNA". 

Line 105. Proper website information for reference genome need to be included.

Line 105, 108. References should be included to the procedures.

Line 116-118. Please include the reference and software packages that were used in this study.

Line 143. Please add the full name of HMGS.

Line 158. Please rephrase "All qRT-PCR gene-specific primers". 

Line 203-204.  Add reference.

Line 218. In Figure 3, it is recommended to include more detailed information, i.e., full name of genes, full name of HW, YW, HN, and YN, Normalizer gene information, statistical analysis, meaning of * and **. 

Line 293, Line 300, Line 302, Line 210. Gene name should be italic.

Line 295, Line 302.  Figure number should be included. Figure 8?

Reviewer 4 Report

In this manuscript, Li et al. report results of a transcriptome analysis of feather follicles on the wing and neck of differently colored chickens. The aim of the study was to identify genetic regulators of melanin deposition in chicken feathers. The authors determined differentially expressed genes and performed GO and KEGG analysis to find links with melanin. Two candidate genes, SLC45A2 and GPNMB, were identified and studied by overexpression in chicken melanocytes, leading to the claim that they promote the proliferation of melanocytes and the deposition of melanin in chickens.

The manuscript is generally interesting. The conclusions are not fully supported by experimental data. It is worrying that the text refers to “hair follicles” more than 10 times, although the authors certainly know that birds to not have hair follicles. This error raises the question as to whether the content of the manuscript was carefully checked by all co-authors.

Figures 2, 4, and 5 are shown at a too small size to reveal necessary details. This presentation is not scientifically valid. Please make these figures bigger.

The legends must be extended to explain what the figures show. This applies especially to Figure 3, but also to other figures.

Lines 295, 302, 310: The number of the figure is missing.

The citations of relevant literature on pigmentation of birds is rather incomplete. Please give a better overview of previous papers and cite more publications.

The quality of English language is good except for misnaming feather follicles "hair follicles".

Round 2

Reviewer 3 Report

Revisions have been made extensively, satisfying the reviewer.

Reviewer 4 Report

Thank your for the corrections and the improvement of the text.